# Past–future information bottleneck for sampling molecular reaction coordinate simultaneously with thermodynamics and kinetics

Yihang Wang [1], João Marcelo Lamim Ribeiro[2] & Pratyush Tiwary[2]

The ability to rapidly learn from high-dimensional data to make reliable bets about the future is crucial in many contexts. This could be a fly avoiding predators, or the retina processing gigabytes of data to guide human actions. In this work we draw parallels between these and the efficient sampling of biomolecules with hundreds of thousands of atoms. For this we use the Predictive Information Bottleneck framework used for the first two problems, and re-formulate it for the sampling of biomolecules, especially when plagued with rare events. Our method uses a deep neural network to learn the minimally complex yet most predictive aspects of a given biomolecular trajectory. This information is used to perform iteratively biased simulations that enhance the sampling and directly obtain associated thermodynamic and kinetic information. We demonstrate the method on two test-pieces, studying processes slower than milliseconds, calculating free energies, kinetics and critical mutations.

[1] Biophysics Program and Institute for Physical Science and Technology, University of Maryland, College Park, MD 20742, USA. [2] Department of Chemistry and Biochemistry and Institute for Physical Science and Technology, University of Maryland, College Park, MD 20742, USA. Correspondence and requests for materials should be addressed to P.T. (email: ptiwary@umd.edu)

A key contributor to the rich and diverse functioning of molecular systems is the presence of myriad possible configurations. Instead of simply staying in the ground state, a given system can adopt one of many metastable configurations and stay trapped there for extended periods. It has been a long-standing dream to apply all-atom molecular dynamics (MD) simulations to learn what these metastable states are, their thermodynamic propensities, pathways for moving between them, and associated kinetic constants. However, there have been two central challenges in this: (a) the large number of states and pathways for traversing between them and (b) the inherent rare event nature of transition between states, wherein a simulation would simply be trapped in whichever metastable state it was started in. While multiple sampling methods[1] and even ultra-specialized supercomputers[2] have been introduced for tackling this timescale problem, the problem is not fully solved. For instance, a large class of sampling methods need an a priori sense of a reaction coordinate (RC), which is a low-dimensional summary of the many configurations and pathways[3–7]. However, this leads to an inherent coupled problem where one needs extensive sampling of rare events to learn the RC, but also needs to know the RC in the first place to perform sampling.

To address this problem, our ansatz is that efficient sampling of energy landscapes of molecular systems has the same key underlying challenge as one faced by a fly as it goes about surviving[8], or the human brain trying to process how to catch a moving baseball[9]. Namely, given limited storage and computing resources, which memories to preserve and which ones to ignore in order to be best prepared for various possible future challenges? This can be paraphrased as the ability to rapidly learn a low-dimensional representation of a complex system that carries maximal information about its future state. Since storing and processing large amounts of information can be computationally and thus energetically expensive for the brain, it has been suggested that neurons in the brain separate predictive information from the non-predictive background in a way that by encoding and processing a minimum amount of relevant information, the brain can still be maximally prepared for future outcomes. The past–future (or predictive) information bottleneck framework introduced and developed in many forms[8–11] involves implementing such neuronal models from an information theoretic basis that can originally be traced back to Shannon's rate distortion theory.

Here we interpret the RC in molecular systems as such a past–future information bottleneck[10]. We develop a sampling method for small biomolecules that, simultaneously and with minimal use of human intuition, esitmates this bottleneck, its thermodynamics, and its kinetics. The central idea is that not all features of the past carry predictive value for the future. A complex model can be made to be very predictive; however, it will often obscure physical interpretability and also end up capturing noise. In order to address this task, we set up an optimization problem and demonstrate how to solve it through the principle of variational inference[12] implemented through deep neural networks. This makes it possible to estimate a predictive information bottleneck (PIB)[11], which we interpret as the RC that, given a molecule's past trajectory, is maximally predictive of its future behavior. Our net product is an iterative framework on the lines of ref. [13] that starts from a short MD simulation, and, given these data, makes an estimate of the RC, its Boltzmann probability, and its associated causal Green's function valid for short times. This information is leveraged to perform systematically biased simulations with enhanced exploration of phase space, which can then be used to re-learn the RC along with its probability and propagator, and iterating between MD and variational inference until optimization is achieved. At this point, we have converged

estimates of the most informative degrees of freedom, associated metastable states, and their equilibrium probabilities. Finally, through the use of a generalized transition state theory-based framework on the lines of ref. [14], we recover the unbiased kinetics for moving between different metastable states.

We first demonstrate the method on sampling a small peptide. We then apply it to a problem of immense relevance, by calculating the full dissociation process of benzene from L99A mutant of the T4 lysozyme protein[15,16]. In the last system, with the use of all-atom MD simulations taking barely a few hundred nanoseconds in total, and with the minimal use of prior human intuition as in other related methods, we obtain accurate thermodynamic and kinetic information for a process that takes few hundred milliseconds in reality. Our simulations shed light on the complex interplay between protein flexibility and ligand movement, and predict the residues whose mutations will have the strongest effect on the ligand dissociation mechanism. We believe our approach marks a big step forward in the use of fully-automated all-atom simulations for the study of complex molecular and biomolecular mechanisms.

## Results

**Principle of past–future information bottleneck.** We formalize this problem in terms of a high-dimensional signal $\mathbf{X}$ characterizing the state of a N-particle system under some generic set of thermodynamic conditions. We take $\mathbf{X}$ as some $d$ generalized coordinates or basis set elements, where $1 \ll d \ll N$. Let the value of this signal measured at time $t$, or the past, be denoted by $\mathbf{X}_t$ and at time $t + \Delta t$, or the future by $\mathbf{X}_{t + \Delta t}$. We call $\Delta t$ the prediction time delay. We assume that $\mathbf{X}_t$ and $\mathbf{X}_{t + \Delta t}$ are jointly distributed as per some probability distribution $P(\mathbf{X}_t, \mathbf{X}_{t + \Delta t})$. The mutual information $I(\mathbf{X}_t, \mathbf{X}_{t + \Delta t})$ (Supplementary Notes 1 and 2 for this and other definitions) quantifies how much an observation at one instant of time $t$ can tell us about an observation at another instant of time $t + \Delta t$. Furthermore, in this article we restrict our attention to stationary systems; hence, we omit the choice of time origin and write down $\mathbf{X}_t$ as $\mathbf{X}$ and $\mathbf{X}_{t + \Delta t}$ as $\mathbf{X}_{\Delta t}$. The principle of PIB[10,11] postulates a bottleneck variable $\chi$, which is related to $\mathbf{X}$ by an encoder function $P(\chi|\mathbf{X})$. Given the bottleneck variable $\chi$, predictions of the future $\mathbf{X}_{\Delta t}$ can be made with a decoder $P(\mathbf{X}_{\Delta t}|\chi)$. PIB says that the optimal bottleneck variable is one which is as simple as possible in terms of the past it needs to know, yet being as powerful as possible in terms of the future it can predict correctly. This intuitive principle can be formally stated through the optimization of an objective function $\mathcal{L}$, which is a difference of two mutual informations:

$$\mathcal{L} \equiv I(\chi, \mathbf{X}_{\Delta t}) - \gamma I(\mathbf{X}, \chi). \tag{1}$$

The above objective function quantifies the trade-off between complexity and prediction through a parameter $\gamma \in [0, \infty)$.

**Variational inference and neural network architecture.** Typically, both the encoder $P(\chi|\mathbf{X})$ and the decoder $P(\mathbf{X}_{\Delta t}|\chi)$ can be implemented by fitting deep neural networks[17] to data in form of time-series of $\mathbf{X}$. Our work stands out in three fundamental ways to typical implementations of the information bottleneck principle and in general of artificial intelligence (AI) methods to sampling biomolecules[18–21]. First, we use a stochastic deep neural network to implement the decoder $P(\mathbf{X}_{\Delta t}|\chi)$, but use a simple deterministic linear encoder $P(\chi|\mathbf{X})$ (see Fig. 1). The simple encoder ensures that the information bottleneck or RC we learn is actually physically interpretable, which is notably hard to achieve in machine learning. On the other hand, by introducing noise in the decoder, we can control the capacity of the model to ensure that the neural network can delineate useful feature from useless

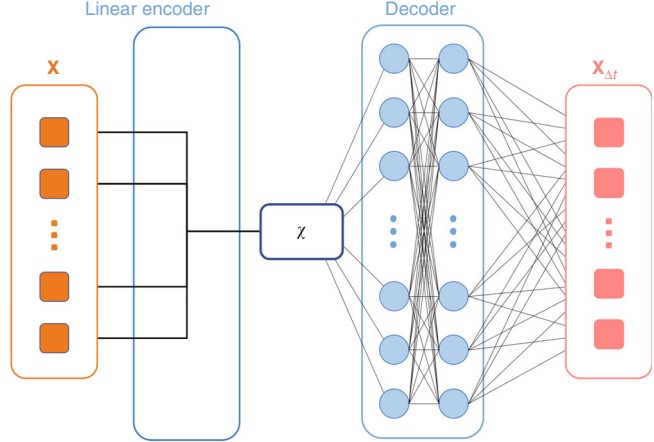

**Fig. 1** Network architecture used for learning predictive information bottleneck $\chi$. The decoder $Q(\mathbf{X}_{\Delta t}|\chi)$ is a stochastic deep neural network, while the encoder $P(\chi|\mathbf{X})$ is of a simple deterministic and thus directly interpretable linear form

information instead of just memorizing the whole dataset. Second, now that our encoder is a simple linear model, we completely drop the complexity term in Eq. (1) and set $\gamma = 0$. Due to a reduced number of variables, this leads to a simpler and more stable optimization problem. Finally, the rare event nature of processes in biomolecules makes it less straightforward to use of information bottleneck/AI methods for enhanced sampling. Here we develop a framework on the lines of ref. [13], which makes it possible to maximize the objective function in Eq. (1) through the use of simulations that are progressively biased using importance sampling as an increasingly accurate information bottleneck variable is learnt.

Our typical starting point is an unbiased MD trajectory $\mathbf{X} = \{\mathbf{X}^1, \ldots, \mathbf{X}^M\}$ with $M$ data points. We want to develop a low-dimensional mapping $\chi$ of this high-dimensional space, which maximizes the objective function $\mathcal{L} = I(\chi(\mathbf{X}), \mathbf{X}_{\Delta t})$. At the heart of this mutual information lies the calculation of the decoder $P(\mathbf{X}_{\Delta t}|\chi)$, which can in principle be done exactly using Bayes' theorem (Supplementary Note 1). However, this becomes impractical as soon as the dimensionality of $\mathbf{X}$ increases, due to a fundamental problem in statistical mechanics and machine learning: intractability of the partition function in high dimensions[22]. The principle of variational inference is an elegant and powerful approach to surmount this problem[23].

Let us consider a generic encoder given by some conditional probability $P_{\theta}(\chi|\mathbf{X})$, where $\theta$ is a set of parameters. Our objective then is to find the optimal RC or equivalently, the encoder $\theta$ that optimizes the PIB objective:

$$\theta^* = \text{argmax}_{\theta}\mathcal{L}(\theta). \qquad (2)$$

As mentioned above, this optimization problem is intractable for almost all cases of practical interest. However, it is possible to perform an approximate inference problem by assuming an approximate decoder $Q_{\phi}(\mathbf{X}_{\Delta t}|\chi)$ parametrized by the vector $\phi$. For any choice of $\phi$, we make a straightforward use of Gibbs' inequality[12] to write down (Supplementary Note 2):

$$\begin{aligned} I(\chi, \mathbf{X}_{\Delta t}) &= H(P_{\theta}(\mathbf{X}_{\Delta t})) - H(P_{\theta}(\mathbf{X}_{\Delta t}|\chi)) \\ &\geq H(P_{\theta}(\mathbf{X}_{\Delta t})) - C(P_{\theta}(\mathbf{X}_{\Delta t}|\chi)||Q_{\phi}(\mathbf{X}_{\Delta t}|\chi)). \end{aligned} \qquad (3)$$

Here $H$ and $C$ denote Shannon entropy and cross entropy, respectively. Take note that the first term in Eq. (3) is independent of our model parameters and hence can be completely ignored from the optimization. Focusing on the second term in Eq. (3), we thus obtain a variational lower bound

on the PIB objective function:

$$\mathcal{L} \geq \mathcal{L}' = -C(P_{\theta}(\mathbf{X}_{\Delta t}|\chi)||Q_{\phi}(\mathbf{X}_{\Delta t}|\chi)). \qquad (4)$$

Thus, $\mathcal{L}'$ is a tractable lower bound bound to the true PIB objective function $\mathcal{L}$, which involves a variational approximation through the trial decoder parametrized by $\phi$. It has a simple physical interpretation. We are attempting to learn a decoder probability function $Q$ that mirrors the actual Bayesian inverse probability function $P$ in terms of predicting the future state $\mathbf{X}_{\Delta t}$ of the system, given the knowledge of the RC $\chi$. The difference between the two probability distributions is calculated as a cross-entropy. By maximizing the right-hand side of Eq. (4) simultaneously with respect to the decoder and encoder parameters $\phi$ and $\theta$, respectively, we can then solve the actual optimization problem posited in Eq. (4) rigorously and identify the optimal RC.

It is clear that a model of a dynamical system $\mathbf{X}$ that attempts to capture just its stationary probability $P(\mathbf{X})$ will be less informative and useful than one that captures the joint past-future probability distribution $P(\mathbf{X}, \mathbf{X}_{\Delta t})$. This is simply because the stationary probability can always be calculated by integrating $P(\mathbf{X}, \mathbf{X}_{\Delta t})$ over future outcomes $\mathbf{X}_{\Delta t}$. What is however less clear is the choice of the time-delay $\Delta t$[20]. In biomolecular systems, it is likely that there will be a hierarchy of time scales and thus time delays relevant to different types of structural and functional details. In principle, our formulation allows us to probe these various time delays in a systematic manner. Here, for the purpose of enhanced sampling, we propose an approach for selecting $\Delta t$ that is rooted in the reactive flux formalism of chemical kinetics[24–26]. This formalism applies to any system with stochastic transitions on a network of microstates with arbitrary, complex connectivity. Summarily, it states that the correlation function for a trajectory's population in any given state can be partitioned into three parts: (a) an initial inertial part, (b) an exponential decay, and (c) an intermediate plateau region between (a) and (b). A key insight from this formalism is that capturing (c), that is, the plateau part of a system's state to state dynamics accurately is necessary and sufficient to capture the temporal evolution at any timescale. By paraphrasing this argument in the context of the present work, we propose to learn our PIB model for gradually increasing values of the predictive time-delay $\Delta t$, and stop when the calculated bottleneck variable converges.

**Variational inference on unbiased and biased trajectories.** We now show how to calculate $\mathcal{L}'$ in practice. For a given unbiased trajectory $\{\mathbf{X}^1, \ldots, \mathbf{X}^{M+k}\}$ with large enough $M$, we can easily show (Supplementary Note 2):

$$\mathcal{L}' = \frac{1}{M}\sum_{n=1}^{M} \log Q(\mathbf{X}^{n+k}|\chi^n), \qquad (5)$$

where $\chi^n$ is sampled from $P(\chi|\mathbf{X^n})$ and the time interval between $\mathbf{X}^n$ and $\mathbf{X}^{n+k}$ is $\Delta t$. For practical rare event systems, however, a typical MD trajectory will be trapped in the state where it was started. Here we use our current best estimate of the PIB to perform importance sampling of the landscape, so that the system is more likely to sample different regions in configuration space, and use this enhanced sampling to iteratively improve the quality of the RC. However, the data so generated is biased per definition, and we need to reweight out the effect of the bias. We suppose that along with the time series $\{\mathbf{X}^1, \ldots, \mathbf{X}^{M+k}\}$, we also have been provided the corresponding time-series for the bias $V$ applied to the system $\{V^1, \ldots, V^{M+k}\}$. We can then use the principle of importance sampling[27] to write our PIB objective function $\mathcal{L}'$ as

follows (Supplementary Note 2):

$$\mathcal{L}' = \left\{ \sum_{n=1}^{M} e^{\beta V^n} \right\}^{-1} \sum_{n=1}^{M} e^{\beta V^n} \log Q(\mathbf{X}^{n+k}|\boldsymbol{\chi}^n), \qquad (6)$$

where $\beta$ is inverse temperature. The above equation is however approximate, as it assumes $P_{\text{biased}}(\mathbf{X}^{n+k}|\boldsymbol{\chi}^n) \approx P_{\text{unbiased}}(\mathbf{X}^{n+k}|\boldsymbol{\chi}^n)$. This is exact as $\Delta t \mapsto 0$, and can be expected to be reasonably valid for small $\Delta t$, where we expect that the system on average would not have diffused too far from its starting position at the beginning of that interval. If the bias varies smoothly enough that its natural variation length scale is smaller or comparable to this diffusion distance, then for small enough $\Delta t$ we can indeed make the aforementioned approximation. This means that we select the smallest possible $\Delta t$ at which the RC estimate plateaus.

**Patching it all**. We now state our complete sampling algorithm, which accomplishes in a seamless manner the identification of the RC together with the sampling of its thermodynamics and kinetics. The first step is to perform an initial round of unbiased MD. This trajectory, expressed in terms of $d$ order parameters $\{s_1, \ldots, s_d\}$ (where $1 \ll d \leq N$), is fed to a deep learning module (Fig. 1). The deep learning module implements the optimization of $\mathcal{L}'$ in Eq. (6) through the use of multi-layer feed-forward neural network for the stochastic decoder $Q$, and a physically interpretable linear map for the deterministic encoder $P$ (Fig. 1). Unlike the decoder, the encoder has no noise term and always maps $\{s_1, s_2, \ldots, s_d\}$ to $\sum_i c_i s_i$, where $\{c_i\}$ denote the weights of different order parameters. We perform this optimization for gradually increasing values of the predictive time-delay $\Delta t$, and estimate RC $\chi$ (given by the values of the weights $c_i$) as seen by the first plateau in terms of when it ceases to depend on choice of $\Delta t$. This value of $\Delta t$ is then kept constant for different rounds of our protocol. At this point, we have an initial estimate of $\chi$ and also its unbiased probability distribution $P^u(\chi)$. These are both used to construct a bias potential $V_{\text{bias}}(\chi)$ for the next iteration of MD:

$$V_{\text{bias}}(\chi) = k_B T \log P^u(\chi), \qquad (7)$$

where $k_B$ is Boltzmann's constant and $T = \frac{1}{k_B \beta}$. With this bias potential added to the original Hamiltonian of the system, we run a biased MD simulation. This explores an increased amount of configuration space since we have applied a bias along our estimated slow degree of freedom, viz. the PIB or the RC. This next round of MD trajectory is again fed to the deep learning module, but this time each data point carries a weight $w = e^{\beta V_{\text{bias}}}$ to compensate for the applied bias. This now identifies an improved RC $\chi$ and its unbiased probability through the use of importance sampling:

$$P^u(\chi) \propto \frac{\langle w \delta(\chi - \chi(t)) \rangle_b}{\langle w \rangle_b} \equiv e^{-\beta F(\chi)}, \qquad (8)$$

where the subscript b denotes sampling under a biased ensemble with weight $w = e^{\beta V_{\text{bias}}}$ and $F(\chi)$ is the free energy along $\chi$. From here, using the bias as $-F(\chi)$ our algorithm can now enter into further iterations of MD–deep learning–MD–... This looping continues until both the RC $\chi$ and the free energy estimate $F(\chi)$ along the RC have converged. We have thus obtained an optimized RC and its Boltzmann probability density, or equivalently the free energy. Through these we can directly demarcate the relevant metastable states and quantify their relative propensities. Furthermore, we can also calculate the transition rates for moving between these metastable states. The central idea is to keep all transition states between the different metastable states, as identified through the RC, devoid of any bias. As we show in examples, this can be easily achieved when implementing Eq. (8), by

ensuring that any barriers in the unbiased probability distribution of the estimated RC are completely bias-free. Once we have done this, we take into account that by virtue of it being the PIB, the RC already encapsulates any relevant, predictive modes in the system. Thus, the hidden barriers, which have invariably been corrupted through the addition of such a bias, do not have any predictive power for the dynamics of the system, and are thus not relevant to the process at hand. This then implies that (i) the biased dynamics preserves the state-to-state sequence one would have seen with unbiased dynamics, and (ii) through the use of a simple time rescaling calculation[14,28] (Supplementary Note 1) we can calculate the acceleration of rates achieved through biased simulations. Finally, we can perform self-consistency checks for the reliability of the rescaled kinetics by analyzing the unbiased lifetimes for robustness with precise choice of biasing protocols (Supplementary Fig. 1). We now demonstrate the use of the PIB framework with two biomolecular case studies, in both of which we simultaneously learn the RC, free energy, and kinetic rate constants. In each case, the RC $\chi$ is constructed as a linear combination $\chi = \sum_i c_i s_i$, where $\{c_i\}$ denote the weights of different pre-selected order parameters $\{s_1, s_2, \ldots, s_d\}$.

**Conformation transitions in a model peptide**. First we consider the well-studied alanine dipeptide system (Fig. 2a). This system, as characterized by its Ramachandran dihedral angles, can exist in different metastable states with varying stabilities and hard-to-cross intermediate barriers. However, due to its small size it serves as a reliable benchmark where we can perform longer than microsecond unbiased MD simulations to benchmark our PIB calculations.

Here we choose $\{\cos \phi, \sin \phi, \cos \psi, \sin \psi\}$ as our order parameters, where $\phi$ and $\psi$ are the backbone dihedral angles. By taking trigonometric functions of dihedral angles, we avoid problems related to periodic boundary conditions. The PIB protocol used here is shown in Fig. 2. In the initial round, we perform a short unbiased MD simulation (see Fig. 2a for trajectory and Supplementary Methods for technical details). As discussed in Methods, the RC is then determined as the linear encoder of the trained neural network with smallest loss function. In Fig. 2b, we show how the weights rapidly converge as functions of predictive time delay and reach a plateau in <2 ps, which we set as $\Delta t$. Figure 2c shows the RC $\chi$ as well as the bias $V(\chi)$ learnt along it to be used in the next round of MD. With this biasing potential, we perform biased MD simulation as shown in Fig. 2d. This trajectory through the use of Eq. (8) leads to a more complicated bias structure as shown in Fig. 2e along with the improved RC $\chi$. Biased simulation with this new RC and bias as shown in Fig. 2f finally leads to escape from the starting metastable state. The final obtained RC is: $\chi = 0.02\cos\phi + 0.97\sin\phi - 0.25\cos\psi - 0.02\sin\psi$. It is known for alanine dipeptide that $\phi$ is more relevant than $\psi$ for capturing the conformational transitions, and our PIB-based RC estimate agrees with that. The shift in weights of order parameters across different rounds (Supplementary Fig. 5) reflects how our iterative scheme finds the optimal RC.

Now that we have achieved back-and-forth motion in terms of the rare event we intended to study, we use this final RC and bias to perform multiple sets of longer simulations with no further refinement of the RC. This yields the free energy surface (defined as $-k_B T \log P(\phi, \psi)$, where $P$ is the unbiased Boltzmann probability) as shown in Fig. 2g. This is in excellent agreement with previously published benchmarks for this system[27]. At the same time, we use the acceleration factor to rescale the biased time back to the unbiased time. In Fig. 2h we show the cumulative distribution functions of the first passage time from the deeper

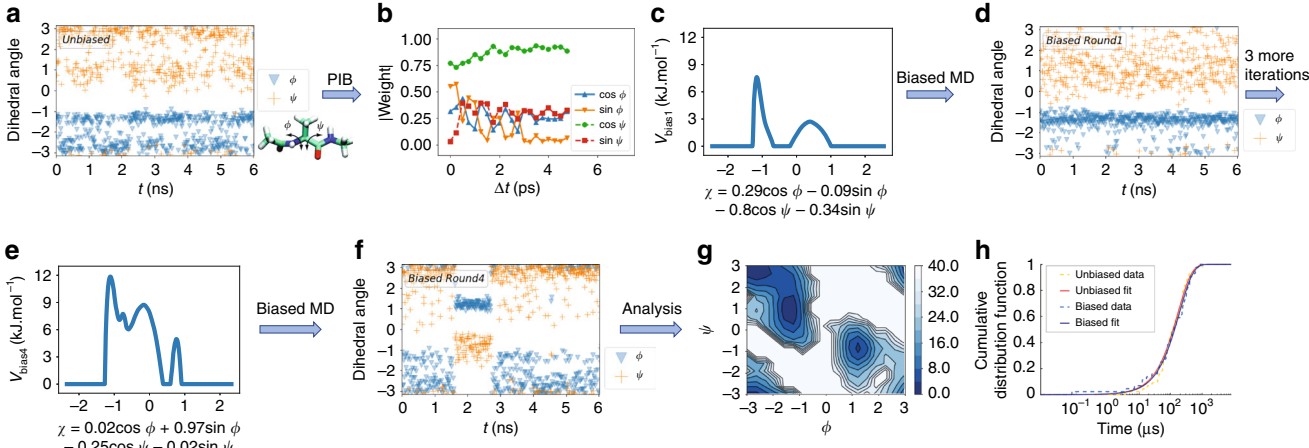

**Fig. 2** Past–future information bottleneck framework results on alanine dipeptide. **a** Unbiased simulation trajectory for dihedral angles $\Phi$ and $\Psi$, in blue triangles and orange pluses, respectively. The alanine dipeptide molecule is shown in inset. **b** Absolute weights for different order parameters in the first training round as a function of the predictive time delay $\Delta t$. Blue triangles, orange triangles, green circles, and red squares correspond to $\cos\phi$, $\sin\phi$, $\cos\psi$, and $\sin\psi$, respectively. **c–f** Free energy along the adaptively learnt reaction coordinate (RC) along with the corresponding biased trajectories for different training rounds. **g** Free energy along $\phi$, $\psi$ after the RC has converged with energy contours every 4 kJ mol$^{-1}$. **h** Kinetics from the post-training biased runs as well as reference unbiased runs. The two are essentially indistinguishable. Orange and blue dashed lines denote unbiased data, respectively, while red and black solid lines show corresponding best fits

basin as obtained through this approach, and through much longer unbiased MD runs, which are feasible given the small size of this system. The distribution functions and their best-fit Poisson curves are nearly indistinguishable, and lead to excellent agreement in values of the escape rate constant, given by $k = 5.2 \pm 0.8$ and $5.8 \pm 0.9 \, \mu\text{s}^{-1}$, respectively, for biased and unbiased simulation.

**Benzene dissociation from T4-L99A lysozyme**. We now apply our framework to a very challenging and important test case, namely the pathway and kinetic rate constant of benzene dissociation from the protein T4-L99A lysozyme in all-atom resolution[15,16]. We also demonstrate how the RC calculated through our approach can be directly used to perform a sensitivity analysis of the protein, and predict the most important residues whose mutations could have a significant affect on the stability of the protein–ligand complex. Such an analysis has direct relevance to predicting, for instance, the mutations in a protein, which could lead to a pharmacological drug losing its efficacy.

For this problem we choose 11 fairly arbitrary order parameters denoted $\{s_1, \ldots, s_{11}\}$. Eight of these are ligand–protein distances, while three are intra-protein distances (Supplementary Fig. 2 and Supplementary Table 1 for order parameter details). The RC is learnt as a linear combination of these order parameters, namely $\chi = \sum_i c_i s_i$.

For this problem as well we start with a short unbiased MD simulation. As shown in Fig. 3, the weights of different order parameters in the RC learnt from this trajectory change as a function of the predictive time-delay $\Delta t$, but converge quickly. On the basis of this plot, we set $\Delta t = 2$ ps for all further calculations. We then iterate—using the same neural network architecture as for alanine dipeptide (Fig. 1)—between rounds of learning an iteratively improved RC $\chi_1$ together with its probability distribution, and running biased MD using the iteration's RC and probability distribution as bias $V_1(\chi_1)$ (Eq. (7)). After nine rounds, we find that the bias saturates as a function of training rounds. That is, no further enhancement in ergodicity is achieved by performing additional rounds of the aforementioned iteration. This corresponds to the system reaching configurations where the

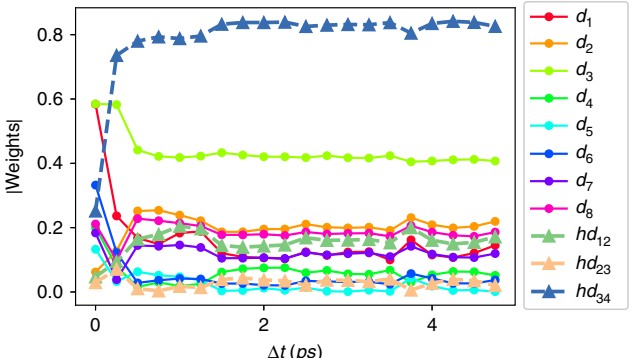

**Fig. 3** Order parameters weights as functions of time delay for benzene–lysozyme dissociation. The scheme shows the absolute value of weights for 11 order parameters in the first round as a function of predictive time delay

previous PIB ceases to be effective. To learn a new PIB, we use the washing out trick from ref. [29] to learn a second RC $\chi_2$ conditioned on our knowledge of the first RC $\chi_1$. In the next few rounds of learning MD iterations, we (a) keep $\chi_1$ and $V_1(\chi_1)$ fixed, and (b) do not account for $V_1(\chi_1)$ when using Eq. (8). Through this we learn a bias $V(\chi_1, \chi_2) = V_1(\chi_1) + V_2(\chi_2)$. In principle we can lift this assumption and learn more complicated non-separable $V(\chi_1, \chi_2)$. In a few rounds of training $\chi_2$, we observed spontaneous disassociation of the ligand from the protein. We are now ready to use the RC $(\chi_1, \chi_2)$ (shown in Fig. 4) and its bias $V(\chi_1, \chi_2)$ learnt to directly study the pathway and kinetics of ligand dissociation.

For this we launch 20 independent biased simulations using $(\chi_1, \chi_2)$ as RC and $V_1(\chi_1) + V_2(\chi_2)$ as bias. By calculating the acceleration factor, we can recover the original timescale of the first passage time. As we show in SI, we fit the cumulative distribution function to a Poisson process and get an escape rate constant of $3.3 \pm 0.8 \, \text{s}^{-1}$, which is in good agreement with other methods[30–32]. We also obtain a range of free energies viewed as functions of different order parameters (Supplementary Fig. 3). These are in excellent agreement with previously published results[29–32].

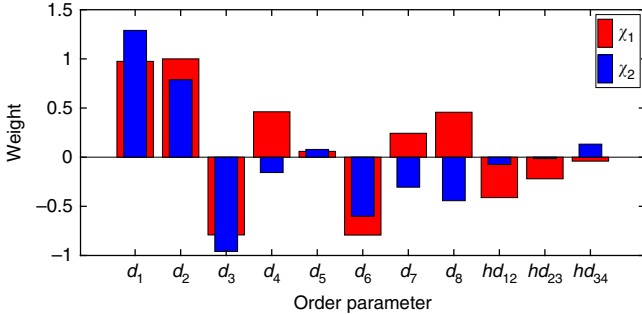

**Fig. 4** The two-component predictive information bottleneck for benzene–lysozyme dissociation, where colors red and blue correspond to $\chi_1$ and $\chi_2$, respectively. The optimized weights for different order parameters are illustrated after scaling all weights to keep $c_1 = 1$ in $\chi_1$

A comment we would like to make here concerns the magnitude of $\Delta t$, especially in connection to the decorrelation time of the MD thermostat. Here we implemented a canonical ensemble using the velocity rescaling thermostat[33] with a time constant of 0.1 ps. Our predictive time-delay $\Delta t$ is thus at least 20 times longer than the times for which the history of the thermostat would persist. Interestingly, as can be seen in Fig. 3 the estimate of the RC converges with longer time delays, which would be even more accurate from the perspective of not having thermostat-induced noise, but would be less accurate due to biasing related errors as explained earlier.

**Predicting critical residues.** On the basis of the PIB that we have now calculated, we can directly predict which protein residues have the most critical effect on the system. To do so, our guiding principle is that the residues which carry higher mutual information with the PIB are more likely to have an impact on the stability of the system, for instance, if these residues were to be mutated. By performing a scan of the mutual information between the PIB and the backbone dihedral angles of different residues, we can rank them as being most critical to least critical (Supplementary Note 3 has further details of the calculation setup). As shown in Fig. 5, some of the important residues are (in order of decreasing relevance): Ser136, Lys135, Asn132, Leu133, Ala134, Phe114, Val57, Asp20, Leu118, and Val131.

These residues can be classified in two broad groups. First, we have group (a) comprising residues 114, 118, and 131–136— together these contribute to breathing movement between the two helices through which the ligand leaves. Second, we have group (b) comprising residues 20 and 57, which lie in different disordered regions of the protein, and have no obvious interpretation. The roles of groups (a) have been hinted at in previous works[32,34,35] and are thus yet another validation of our approach. The role of group (b) during the unbinding process remains to be seen. In order to demonstrate the robustness of this calculation, we performed the full MD–deep learning–MD–deep learning–… iterative protocol with a new set of order parameters (details in Supplementary Discussion) that considered new protein–ligand distances and completely excluded any protein–protein distances, as the selection of latter require a more significant role of human intuition in anticipating protein breathing for instance. In this new set of calculation, we again obtained same critical residues, including a residue from the same disordered region as in group (b) above. Whether the disordered regions are biophysically relevant to the unbinding of the ligand with the existence of a long-range allosteric communication pathway, or if these residues are picked due to just noise from our calculations, needs more detailed mutagenesis study in the future.

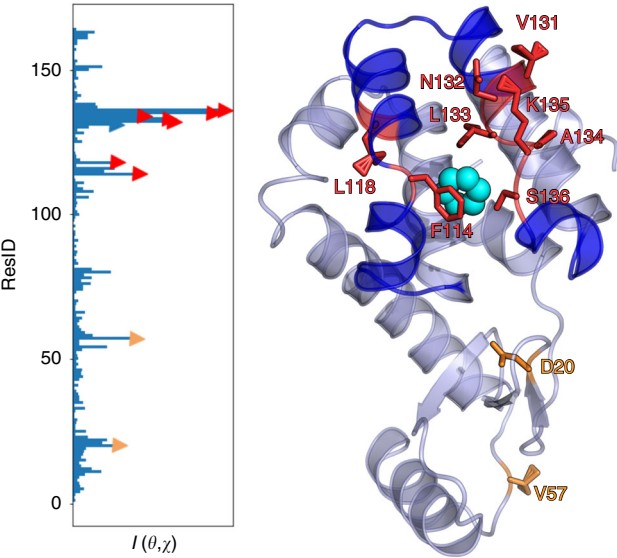

**Fig. 5** Critical residue analysis for benzene–lysozyme complex. The plot on left shows for every residue the maximal mutual information between the predictive information bottleneck (PIB) and either of the Ramachandran angles $\phi$, $\psi$ of that residue. The top 10 residues are highlighted through markers and in the right plot, illustrated relative to the ligand in a typical intermediate pose

## Discussion

In this work we have introduced a new framework for the simultaneous sampling of the RC, free energy and rate constants in biomolecules with rare events. Our work is grounded in the PIB framework, which is an information theoretic approach for building minimally complex yet maximally predictive models from data. Such a framework has previously been found useful for modeling fruit fly movement and human vision. Here we exploit the commonality between these diverse problems and that of sampling complex biomolecular systems, namely the need to quickly predict the future state of a system given noisy and high-dimensional information. Our method implements this framework through the use of a unique linear encoder–stochastic decoder model, where the latter is a deep neural network with inbuilt noise. Here we demonstrated the applicability of the method by studying conformational transitions in a model peptide in vacuum and ligand dissociation from a protein in explicit water, with both systems in all-atom resolution. Through extremely short and computationally cheap simulations, we obtained thermodynamic and kinetic observables for slow biomolecular processes in excellent agreement with other methods, experiments and long unbiased MD. Last but not the least, by virtue of having captured the most predictive degrees of freedom in the system, we could also make, arguably for the first time, direct predictions of how protein sequence can impact dissociation dynamics—namely, which mutations in the protein would be most deleterious to the dissociation process.

We would also like to discuss here some obvious limitations of our approach. The RC learnt from AI can often lack interpretability. We address this issue here through the use of a simple linear encoder, which preserves the interpretability of the RC. However, this comes at the cost of using smartly designed non-linear basis functions, or order parameters, which can often be domain-dependent. For example, different classes of basis functions were needed here for alanine dipeptide conformation change (namely, torsions) and ligand dissociation (namely, distances). Using a linear encoder on distances for alanine dipeptide, or torsions for ligand dissociation, leads to no discernible

enhancement in sampling as we iterate through rounds of deep learning and MD. Thus, bad choices of basis functions for the protocol can be ruled out at least in a heuristic manner by quantifying whether these led to more ergodic sampling or not. We are cautiously optimistic about the applicability of this work to even more complex systems than the ones considered here. In addition to identifying better basis functions, one other issue that we will need to address carefully for such systems is when one or even two RCs are simply not sufficient to describe the process of interest. Often these different coordinates can be entangled, and we might have to use further deep learning machinery in order to deal with such issues[36]. Overall, we believe this work marks an important step ahead in computer simulation of molecules, and should be useful to different communities for robust, reliable studies of rare events.

## Methods

**MD simulations.** The simulations were performed with the software GROMACS 5.0[37,38], patched with PLUMED 2.4[39]. For alanine dipeptide, the temperature was kept constant at 300 K using the velocity rescaling thermostat[33]. The L99A T4L-benzene simulations were done with the constant number, pressure, temperature (NPT) ensemble with temperature 298 K and pressure 1.0 bar. Constant pressure was maintained using Parrinello–Rahaman barostat[40]. The integration time step was 2 fs and order parameters were saved every time step. In each round, four independent simulations with different initial randomized velocities as per Maxwell–Boltzmann distribution were performed to improve the quality of the free energy sampling. Further details about system setup can be found in Supplementary Methods.

**Neural network architecture.** A densely connected layer without activation function was used to linearly encode the order parameters $\mathbf{X}$ to RCs $\chi$. Gaussian noise was added to $\chi$ before being passed to the decoder. Decoder consisted of two hidden layers and an output layer, which were all densely connected. Exponential linear unit was used as the activation function for hidden layers. We assume $Q_{\phi}(\mathbf{X}_{\Delta t}|\chi) = \mathcal{N}(\mathbf{X}_{\Delta t}; f_{\phi}(\chi), \sigma^2)$. $f_{\phi}(\chi)$ corresponds to the decoder part of the neural network, which maps states on the RC to states in order parameter space. With this assumption, maximizing the objective function is equivalent to minimizing the mean square error between $\mathbf{X}_{\Delta t}$ and network prediction $f_{\phi}(\chi)$.

**Neural network hyper-parameters.** Hyper-parameters in this work included the variance of Gaussian noise, the number of neurons in hidden layers, initializer of weights of each layer, and the learning rate for the RMSprop algorithm[23]. In our two case studies, all these hyper-parameters are set to be the same. The variance of Gaussians was kept 0.005. Each hidden layer had 128 neurons. The leaning rate was set to be 0.003. Initial weights of each layer were randomly picked from a uniform within range [−0.005, 005]. The transferability of hyper-parameters between different systems without much tuning reflects that this method is not very sensitive to the choice of neural network hyper-parameters. From our experience, we suggest that the choice of the variance of Gaussian should not be too big as it will wash out meaningful features. For more complicated systems, a deeper (more layers) or wider (more neurons in each layer) decoder might be needed. The tuning of learning rate was done by looking at how order parameter weights change during the training process. If the learning rate was too high, order parameter weights did not converge.

**Neural network training.** Independent simulations often explored different configurations due to the finite and small simulation time. We considered the trajectory with the highest variance to have maximum ergodic exploration and used it to train the RC for the next round. Similar to other non-convex optimization problems, the results could have converged to a local minimum or even a saddle point. To safeguard against such spurious solutions learnt by the neural network, we performed independent training runs with random initial weights in layers. The RC was then determined as the linear encoder of the trained neural network with smallest loss function. We will be refining this procedure in future work as strictly speaking a lower loss function is no guarantee of reaching a better solution.

**Construction of bias.** Trajectories from four independent simulations were mixed to calculate the biasing potential. The minimum bias was set to 0. The maximum of the bias is determined as the supremum of all possible value that satisfied two criteria: (a) the value should be smaller than the summation of maximum bias in the last round and a preset constant $\Delta V$ corresponding to a confidence parameter for poorly sampled regions; (b) no bias is added on identifiable transition states. By doing (a) we can exclude the regions that are poorly sampled. By doing (b) we can calculate the rates of transition from a biased MD.

## Data availability

All the data and GROMACS/PLUMED input files required to reproduce the results reported in this paper are available on PLUMED-NEST (www.plumed-nest.org), the public repository of the PLUMED consortium[41], as plumID:19.028″. Further data including all-atom coordinates can be obtained from the authors upon request.

## Code availability

The code associated with this work is available from the corresponding author on request.

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

## Acknowledgements

We thank Deepthought2, MARCC, and XSEDE (projects CHE180007P and CHE180027P) for computational resources used in this work. P.T. would like to thank the University of Maryland Graduate School for financial support through the Research and Scholarship Award (RASA). Y.W. would like to thank NCI-UMD Partnership for Integrative Cancer Research for finanical support.

## Author contributions

P.T. and Y.W. designed research; P.T., Y.W., and J.M.L.R. performed research; Y.W. analyzed data; Y.W. and P.T. wrote the paper.

## Additional information

**Competing interests:** The authors declare no competing interests.

