## [Peer Review File · Nature Communications]

Reviewers' comments:

Reviewer #1 (Remarks to the Author):

The article of Wang et al reports on an innovative computational approach based on Predictive information bottleneck and biased MD simulations to sample rare events in biological molecules. A complex reaction coordinate, which is traditionally based in this kind of simulations mostly on human intuition, is obtained by short MD runs and it is applied to biased MD simulations, which is then re-used to estimate the reaction coordinate, until when optimization is achieved. The method appears to be powerful and the results on the two standard test cases to which it has been applied in this paper are in agreement with previous simulations.

I have one main comment on the general applicability of the method. The test cases reported in the paper are relatively simple, with respect to standard biological processes such as protein folding or large conformational transitions underlying the functions of complex biological macromolecules of bigger size. These latter are rather standard to simulate nowadays, but also, in some cases, out of reach for biased simulations due to their large size. In the second example in which the benzene ligand has to be dissociated from the lysozyme 20 independent biased simulations are performed, which is in my opinion a large number and would prevent the application of this method to more complex system. I believe that in order to demonstrate that this method represents an important 'step ahead in computer simulations of biomolecules', as stated in the conclusion, the authors should either extend and/or discuss its application to a more complex test case.

Overall I think the article may deserve publication in Nature Communications after my comment has been addressed.

Reviewer #2 (Remarks to the Author):

The manuscript entitled 'Past-future information bottleneck framework for sampling molecular reaction coordinate, thermodynamics and kinetics' by Tiwary and colleagues describes a new method for performing efficient atomistic simulations for large and complex biomolecules. The approach is based on the predictive information bottleneck framework which allows building minimally complex but predictive models for data. This allows constructing optimal reaction coordinates (RC) from a given dataset by employing an autoencoder neural network (NN) containing a deterministic linear encoder and a stochastic decoder. Then different thermodynamic and kinetic observables can be computed with noticeably reduced computational costs, by sampling the RC.

The article is well written, the work should have a significant impact on the large community of researchers working with simulations of biomolecules or other complex high-dimensional systems, and the demonstrated results are quite convincing. Nevertheless, I have the following two significant questions which should be clarified in the article before further consideration:

1. How the canonical nature of the MD simulations used as the input data affect the method? Indeed, the proposed approach is based on learning the relation between the configurations of a molecule at different moments in time. Within microcanonical ensemble, this relation is defined by the equation of motion. In contrast, for the canonical ensemble (which is typically used in the constant temperature MD simulations), the presence of a thermostat leads to decorrelation after a certain time, and the history depends on the thermostat. Hence, there is no straightforward correlation between the configuration of a molecule at the moment t and $t+dt$ whenever dt is large enough. The time shift used in this work is of an order of picoseconds which is comparable to the internal times of conventional thermostats. How does this fact affect the proposed method?

2. The authors claim that the proposed method does not depend much upon a human intuition. At the same time, the choice of the representation of the input data (order parameters or basis functions) indeed depends on human intuition. This probably would not be critical for building the RC in the case of a nonlinear encoder, but the authors restrict themselves to simple linear encoders. Although the reasoning behind this choice is quite understandable, it is not clear how the different choices of the order parameters would affect the final results. For instance, what will happen in the case of the simulations for alanine dipeptide if instead of one of the important dihedral angles we will choose some distance? How in the linear encoder can we combine different types of order parameters? Hence, adding to the publication the comparison of the results for the alanine dipeptide obtained with different sets of the order parameters will be very beneficial.

We thank this reviewer for the very positive report. Below we provide our response in bold (after double asterisks **) to the comment of this reviewer (in italics). All changes in the manuscript are indicated in the color red.

The article of Wang et al reports on an innovative computational approach based on Predictive information bottleneck and biased MD simulations to sample rare events in biological molecules.

A complex reaction coordinated, which is traditionally based in this kind of simulations mostly on human intuition, is obtained by short MD runs and it is applied to biased MD simulations, which is then re-used to estimate the reaction coordinate, until when optimization is achieved.

The method appears to be powerful and the results on the two standard test cases to which it has been applied in this paper are in agreement with previous simulations.

I have one main comment on the general applicability of the method. The test cases reported in the paper are relatively simple, with respect to standard biological processes such as protein folding or large conformational transitions underlying the functions of complex biological macromolecules of bigger size. These latter are rather standard to simulate nowadays, but also, in some cases, out of reach for biased simulations due to their large size. In the second example in which the benzene ligand has to be dissociated from the lysozyme 20 independent biased simulations are performed, which is in my opinion a large number and would prevent the application of this method to more complex system. I believe that in order to demonstrate that this method represents an important 'step ahead in computer simulations of biomolecules', as stated in the conclusion, the authors should either extend and/or discuss its application to a more complex test case.

Overall I think the article may deserve publication in Nature Communications after my comment has been addressed.

**** This is a very good point and we have now added a discussion to this effect in the main text. We would like to emphasize to this referee that the need to perform 20 independent simulations came not due to our method per se but due to the stochastic nature of ligand**

dissociation, where obtaining accurate residence time essentially means sampling a large number of events. If we had just restricted ourselves to 1 simulation, we would have sampled one of the many dissociation events, without obtaining information on its mean dissociation time as well as the associated errors.

Regarding the referee's comments on more complex systems, we fully agree with the sentiment of this referee and choose to be cautiously optimistic. In this connection, the following text has been added to the discussion section:

"We are cautiously optimistic about the applicability of this work to even more complex systems than the ones considered here. In addition to identifying better basis functions, one other issue that we will need to address carefully for such systems is when one or even two reaction coordinates are simply not sufficient to describe the process of interest. Often these different coordinates can be entangled, and we might have to use further deep learning machinery in order to deal with such issues."

We thank this reviewer for the very positive report. Below we provide our response in bold (after double asterisks **) to the two comments of this reviewer (in italics). All changes in the manuscript are indicated in the color blue.

The manuscript entitled 'Past-future information bottleneck framework for sampling molecular reaction coordinate, thermodynamics and kinetics' by Tiwary and colleagues describes a new method for performing efficient atomistic simulations for large and complex biomolecules. The approach is based on the predictive information bottleneck framework which allows building minimally complex but predictive models for data. This allows constructing optimal reaction coordinates (RC) from a given dataset by employing an autoencoder neural network (NN) containing a deterministic linear encoder and a stochastic decoder. Then different thermodynamic and kinetic observables can be computed with noticeably reduced computational costs, by sampling the RC.

The article is well written, the work should have a significant impact on the large community of researchers working with simulations of biomolecules or other complex high-dimensional systems, and the demonstrated results are quite convincing. Nevertheless, I have the following two significant questions which should be clarified in the article before further consideration:

1. How the canonical nature of the MD simulations used as the input data affect the method? Indeed, the proposed approach is based on learning the relation between the configurations of a molecule at different moments in time. Within microcanonical ensemble, this relation is defined by the equation of motion. In contrast, for the canonical ensemble (which is typically used in the constant temperature MD simulations), the presence of a thermostat leads to decorrelation after a certain time, and the history depends on the thermostat. Hence, there is no straightforward correlation between the configuration of a molecule at the moment t and $t+dt$ whenever dt is large enough. The time shift used in this work is of an order of picoseconds which is comparable to the internal times of conventional thermostats. How does this fact affect the proposed method?

**** This is a very good point and we have now added a discussion to this effect in the main text. The following text has been added:**
“A comment we would like to make here concerns the magnitude of Δt especially in connection to the decorrelation time of the MD thermostat. Here we implemented a canonical ensemble using the velocity rescaling thermostat (33) with a standard time-constant of 0.1 ps. Our predictive time-delay Δt is thus at least 20 times longer than the times for which the history of the thermostat would still persist. Interestingly, as can be seen in Fig. 3 the estimate of the RC converges with longer time-delays which would be even more accurate from the perspective of not having thermostat-induced noise, but would be less accurate due to biasing related errors as explained in the Theory section.”

2. The authors claim that the proposed method does not depend much upon a human intuition. At the same time, the choice of the representation of the input data (order parameters or basis functions) indeed depends on human intuition. This probably would not be critical for building the RC in the case of a nonlinear encoder, but the authors restrict themselves to simple linear encoders. Although the reasoning behind this choice is quite understandable, it is not clear how the different choices of the order parameters would affect the final results. For instance, what will happen in the case of the simulations for alanine dipeptide if instead of one of the important dihedral angles we will choose some distance? How in the linear encoder can we combine different types of order parameters? Hence, adding to the publication the comparison of the results for the alanine dipeptide obtained with different sets of the order parameters will be very beneficial.

**** This is also an important point and we have now added a discussion to this effect in the main text. The following text has been added:**

“We would also like to discuss here some obvious limitations of our approach. The RC learnt from AI can often lack interpretability. We address this issue here through the use of a simple linear encoder which preserves the interpretability of the RC. However this comes at the cost of using “smart” non-linear basis functions, or order parameters, which can often be domain-dependent. For example, different classes of basis functions were needed here for alanine

dipeptide conformation change (namely, torsions) and ligand dissociation (namely, distances). Using a linear encoder on distances for alanine dipeptide, or torsions for ligand dissociation, leads to no discernible enhancement in sampling as we iterate through rounds of deep learning and MD. Thus, bad choices of basis functions for the protocol can be ruled out at least in a heuristic manner by quantifying whether these led to more ergodic sampling or not.”

Reviewers' comments:

Reviewer #1 (Remarks to the Author):

The authors have replied to my concerns. The article can now be published.

Reviewer #2 (Remarks to the Author):

While all of the details of the employed method are well described, and the presented results demonstrate the efficiency of the developed approach, the dependence of the technique upon the human choice of the molecular descriptor is not clear. The text added to the revised manuscript sheds little light on this issue: "We would also like to discuss here some obvious limitations of our approach. The RC learnt from AI can often lack interpretability. We address this issue here through the use of a simple linear encoder which preserves the interpretability of the RC. However this comes at the cost of using "smart" non-linear basis functions, or order parameters, which can often be domain-dependent. For example, different classes of basis functions were needed here for alanine dipeptide conformation change (namely, torsions) and ligand dissociation (namely, distances). Using a linear encoder on distances for alanine dipeptide, or torsions for ligand dissociation, leads to no discernible enhancement in sampling as we iterate through rounds of deep learning and MD. Thus, bad choices of basis functions for the protocol can be ruled out at least in a heuristic manner by quantifying whether these led to more ergodic sampling or not."

Unfortunately, it does not answer the main question – how to choose "good" basis functions? And, assuming that there are multiple choices of "good" order parameters, it is not clear how a particular choice of them will affect the final results including the obtained critical residues? Hence, I have to repeat my previous comment insisting that adding to the publication the comparison of the results for the same molecule but obtained with different sets of the order parameters is essential. If the authors can show that their results are not just a consequence of a lucky choice of order parameters, then the method is a valuable tool for a broad audience of chemists, physicists, and biologists and worth publishing in Nature Communications. Otherwise, the method is highly dependable on a human intuition of a particular researcher and its applications to complex systems is questionable.

Summarizing, I cannot recommend the publication of the manuscript in its current state. At the same time, I fully agree with the authors that their work can be a valuable contribution to the field of simulations and study of complex molecules and I strongly encourage them to do the necessary additional work to revise their results as discussed above.

We thank this reviewer for the overall supportive report and for suggesting additional calculations, which we now have performed, and which have helped us improve the quality of this work. Below we provide our response in bold (after double asterisks **) to the comment of this reviewer (in italics). All changes in the manuscript are indicated in the color blue.

1. While all of the details of the employed method are well described, and the presented results demonstrate the efficiency of the developed approach, the dependence of the technique upon the human choice of the molecular descriptor is not clear. The text added to the revised manuscript sheds little light on this issue: “We would also like to discuss here some obvious limitations of our approach. The RC learnt from AI can often lack interpretability. We address this issue here through the use of a simple linear encoder which preserves the interpretability of the RC. However this comes at the cost of using “smart” non-linear basis functions, or order parameters, which can often be domain-dependent. For example, different classes of basis functions were needed here for alanine dipeptide conformation change (namely, torsions) and ligand dissociation (namely, distances). Using a linear encoder on distances for alanine dipeptide, or torsions for ligand dissociation, leads to no discernible enhancement in sampling as we iterate through rounds of deep learning and MD. Thus, bad choices of basis functions for the protocol can be ruled out at least in a heuristic manner by quantifying whether these led to more ergodic sampling or not.”

Unfortunately, it does not answer the main question – how to choose “good” basis functions? And, assuming that there are multiple choices of “good” order parameters, it is not clear how a particular choice of them will affect the final results including the obtained critical residues? Hence, I have to repeat my previous comment insisting that adding to the publication the comparison of the results for the same molecule but obtained with different sets of the order parameters is essential. If the authors can show that their results are not just a consequence of a lucky choice of order parameters, then the method is a valuable tool for a broad audience of chemists, physicists, and biologists and worth publishing in Nature Communications. Otherwise, the method is highly dependable on a human intuition of a particular researcher and its applications to complex systems is questionable.

Summarizing, I cannot recommend the publication of the manuscript in its current state. At the same time, I fully agree with the authors that their work can be a

valuable contribution to the field of simulations and study of complex molecules and I strongly encourage them to do the necessary additional work to revise their results as discussed above.

**** This is a very good point and we agree that it is important to show that our result is not a consequence of a lucky choice of order parameters. As suggested by the referee, we used an entirely new set of arbitrarily chosen order parameters to study the L99A T4L-benzene system and explored if the critical residue calculation is indeed robust. In this new set, we removed all protein-protein distances as it might be deemed to require more intuition or prior knowledge of the system, compared to using only protein-ligand distances. The only order parameters left in the set are new ligand-protein distances which should be easier to come up with, as it is natural to use ligand-protein distance to describe the position of ligand relative to the binding pocket. Our results show that reversible ligand-unbinding can still be achieved with this new set of order parameters and critical residues calculation picked up similar residues as our previous results. We would like to emphasize that our method is not totally intuition free and in most cases, indeed adding some "high quality" order parameters by human intuition can improve the performance of this method. But we believe that perfect intuition in choosing order parameters is neither necessary nor sufficient for the success of this method, and our method seems to work well with wide range of choices, as now demonstrated through concrete examples.**

The following text has been added to the main text along with a new section and figures in the SI (all marked in color blue):

" These residues can be classified in two broad groups. Firstly, we have group (a) comprising residues 114, 118, and 131--136 - together these contribute to breathing movement between the two helices through which the ligand leaves. Secondly we have group (b) comprising residues 20 and 57, which lie in different disordered regions between ordered parts of the protein, and have no obvious interpretation. The roles of groups (a) have been hinted at in previous works and are thus yet another validation of our approach. The role of group (b) during the unbinding process remains to be seen. In order to demonstrate the robustness of this critical residue calculation, we

performed the full MD--deep learning--MD--deep learning--... iterative protocol with a new set of order parameters (details in SI) that considered new protein-ligand distances and completely excluded any protein-protein distances, as the selection of latter require a more significant role of human intuition in anticipating protein breathing for instance. In this new set of calculation, we again obtained same critical residues, including a residue from the same disordered region as in group (b) above. Whether the disordered regions are biophysically relevant to the unbinding of the ligand with the existence of a long-range allosteric communication pathway, or if these residues are picked due to just noise from our calculations, needs more detailed mutagenesis study in the future. ”

REVIEWERS' COMMENTS:

Reviewer #2 (Remarks to the Author):

I am satisfied with the revised manuscript. The authors have introduced changes and discussion that make it clearer for the reader the potential applications and limitations of the presented approach. I am happy to recommend publication.